# Optimal Control Strategy of a Sewer Network

Iulian Vasiliev [1,2,*], Laurentiu Luca [1], Marian Barbu [1] , Ramon Vilanova [2] and Sergiu Caraman [1]

1   Department of Automation and Electrical Engineering, Dunarea de Jos University of Galati,
    Domneasca No. 47, 800008 Galati, Romania; laurentiu.luca@ugal.ro (L.L.); marian.barbu@ugal.ro (M.B.);
    sergiu.caraman@ugal.ro (S.C.)
2   Department of Automation Systems and Advanced Control Research, Autonomous University of Barcelona,
    08193 Barcelona, Spain; ramon.vilanova@uab.cat
*   Correspondence: iulian.vasiliev@ugal.ro

**Abstract:** This paper proposes a series of methods to increase the efficiency of the operating of a sewer network that serves a medium-sized city with a population of 250,000 inhabitants. The sewer network serves five areas of the city and consists of seven tanks that communicate with one another and with the treatment plant through pipes. The controls are applied to the process by valves and pumps. The main objective of this paper is to determine the optimal controls to minimize two performance criteria: volume of overflow, and overflow quality index. The sewer network was modeled in the BSMSewer environment. The optimization of the operating of the sewer network was carried out in the conditions of an influent computed in relation to the number of inhabitants and to the area served, using genetic algorithms as a method of optimization. Five optimization strategies were analyzed by numerical simulation. The analysis of the five strategies was done by comparison of their results with one another, as well as in relation to the case where all of the controls were set at maximum values of 100%. The simulations showed that the third strategy produced the best results in relation to each of the two criteria.

**Keywords:** sewer network; wastewater; control strategy; optimization; genetic algorithm





## 1. Introduction

The development of human communities has led to an increase in the volume of polluted water that must be treated and discharged into natural receptors (rivers, lakes, groundwater etc.). On average, a person uses between 79 and 307 L of water every day for drinking, cooking, washing, cleaning, and hygiene. Naturally, this water used must be collected and treated before returning to the environment. Industrial activities also produce wastewater that can endanger the environment when discharged. This has led to the adoption of strict environmental protection legislation regarding the volume and the quality of water discharged into natural receptors. Currently, the issue of collecting and treating polluted water is treated in an integrated way, as clearly set forth by the European Water Quality Directive adopted by the European Union in 2000.

The main issues that arise regarding wastewater are as follows:

1.  The wastewater collection from various domestic or industrial activities and/or stormwater. It should be noted that in case of large rainfall events and significant amounts of stormwater, it is possible to overload water collection systems, leading to discharges of the excess water into the environment—discharges that can cause flooding or lead to soil and natural water pollution;
2.  The wastewater treatment in treatment plants, after which it can be discharged into natural receptors.

The wastewater collection is carried out within the sewer networks, which consist of pipes, tanks, pumps, etc. Several factors influence the design of a sewer network, including ground conditions, topography, effluent composition, and construction sequencing. These

factors determine whether networks are transverse, perpendicular, parallel, zonal, or radial. Wastewater that flows through the sewer system can be transported either through gravity flow or by implementing pumping stations in situations where gravity flow is not possible. Next, the wastewater is transported to the treatment plants, where the objective is to obtain an effluent that complies with the legislation in terms of its loading with pollutants before discharge.

Current wastewater collection systems are of two types [1]: (1) Combined systems that collect both wastewater from human activity and stormwater, which is then transported to the wastewater treatment plants. Such wastewater collection systems are very sensitive to the quantities of stormwater they collect, with the water transported through them significantly changing both their flow and quality during the rainfall events; (2) separate systems—in these systems only the wastewater is transported to the treatment plant, with untreated stormwater being discharged into natural receptors.

Regarding the modeling of the wastewater collection systems, detailed models have been adopted that describe the phenomena that take place in the sewer pipes in depth, as well as simplified models that can be used in control algorithms [2,3]. The most complex models are the hydrodynamic ones, which are based on Saint-Venant-type equations for one-dimensional non-stationary flow. Other models used are those based on hydrological models, in which the pipes are described by first-order differential equations or those described in [4], where the sewer network is modeled as a hybrid system containing both subsystems, with continuous states as well as discrete states.

The authors of [5] present a dynamic model that describes the reactions of the compounds in the sewer pipes. The purpose of the study was to analyze the reaction rates involved in these reactions (the growth rate of microorganisms, feeding rate, consumption rate, oxygen supply rate, etc.).

In [6], an analysis of the pollution level of discharged water during pluviometric events is made. A number of variables are considered (the maximum turbidity, the total event rainfall, the previous dry weather period, etc.), with the relations between these variables being defined through a statistical model. The authors of [7] propose a simplified semi-distributed urban water quality model, EmiStatR, which brings uncertainty and sensitivity analyses of urban drainage water quality models within reach of practitioners.

In [8], the authors use "deep learning" methods to predict water quality in urban sewer networks. Water quality analysis and prediction is based on the following indicators: environmental indicators (such as area and diameter); social indicators (such as population); water quantity indicators (such as drinking water supply, wastewater flow, water velocity, and liquid level), and water quality indicators that are easy to monitor (e.g., pH, temperature, and conductivity).

The authors of [9] analyzed how the measurement and the life cycle of an integrated system (wastewater treatment plant (WWTP) and combined sewer system) could be influenced by rainfall. This study was based on correlation analyses and regression models to assess the impact of precipitation on the environmental quality.

In [10], a benchmark simulation model (BSMSewer) is described in order to evaluate control strategies for the urban catchment and sewer network. It contains various modules describing wastewater generation in the catchment, along with its subsequent transport and storage in the sewer system. The authors of [11] present a library of dynamic modeling tools for estimating micropollutant fluxes within an integrated urban wastewater and stormwater system (including a drainage network, stormwater treatment units, wastewater treatment plants, sludge treatment, and the receiving water body). In [12], detailed models of sewers, treatment plants, and receiving waters were created in order to describe the performances according to the individual needs and objectives.

The literature reports various control strategies. A flow control system that operates on an existing sewer system is presented in [13]. The practically implemented method is model-based control. Online radar data and radar prognosis are used to model the input flow into the sewer system. Based on these data, the model computes the optimized

set-points for flow control at 12 stormwater retention tanks in the sewer system. The system's operation significantly reduces the amount of rainwater discharged into rivers and water bodies, and more rainwater is treated in the wastewater treatment plant. The authors of [14] present an automatic control strategy (equal filling degree) that can be applied to any type of sewer network, regardless of its size. This method can be a starting point for additional control (e.g., by optimization or manual control). In [15] a method of water pollution control is presented that has the drawback of the need for a large-capacity storage tank. The control method (water-quality-based double-gates control strategy for combined sewer overflows pollution control) uses the chemical oxygen demand (COD) as a control criterion. The control method gives good results even in rainfall conditions. In [16], neural networks were used to predict the discharges in the sewer systems. Different neural networks were investigated using different sets of input and output data, resulting in the neural network with 50 neurons giving the best performance.

In [17], a three-level hierarchical control structure for municipal wastewater systems is proposed; the first level contains sensors and actuators connected to the PLCs, the second level consists of advanced process control systems for wastewater treatment plants and sewer systems, and the third is dedicated to the administrative level. The authors of [18] present another three-level control structure: The first level is the adaptation level, in which the prediction of the network input flows based on the precipitation measurements takes place. The second level is the optimization level that calculates the trajectories for the state and control variables for the entire sewer network, and the third control level is the distributed control level, where the controls calculated by level 2 represent set-points for the decentralized control of each tank.

The issue of controlling the wastewater collection networks has been addressed in many cases by using different optimization algorithms to avoid discharges, efficient use of the treatment plant, and efficient use of the storage tank network [3,18,19]. In [20], a quick pluvial flood warning system using rainfall thresholds is proposed. A tabu search algorithm is implemented with initial boundary conditions based on hydrological analysis in order to optimize the rainfall thresholds. The choice of the optimization method depends on the system's characteristics, the available data, and the objectives of the control strategy. Thus, in [21], linear programming was used to control the storage capacity in a large wastewater collection system. Dynamic programming has been used to design a minimum cost sewer network [22] and to control the San Francisco sewer system [23]. The theory of squared linear control has been used in the multivariable linear control of wastewater collection systems [24,25]. Last, but not least, artificial intelligence, the Internet of things, and genetic algorithms have been used [19,26,27] for the design of wastewater collection systems and the control of their discharges into natural receptors, as well as multi-agent systems in optimization problems (e.g., negotiation mechanisms between various polluting entities for their access to the wastewater treatment plant) [28].

In this paper, the authors propose five optimization strategies based on genetic algorithms of a sewer network for a medium-sized city in Romania with 250,000 inhabitants. The network was modeled in the BSMSewer environment.

The rest of this paper is structured as follows: Section 2 presents the structure and the characteristics of the sewer network, the influent provided by the five areas of the city, the performance criteria, and a brief description of the optimization method; Section 3 shows the simulation results obtained with the five optimization strategies, and the last two sections are dedicated to the discussions and conclusions.

## 2. Materials and Methods

### 2.1. Structure and Characteristics of the Sewer Network

As stated above, a sewer network serving a medium-sized city with a population of 250,000 inhabitants was considered as a case study. The structure of this network is presented in detail in [29]. It contains 7 tanks, as shown in Figure 1, noted as $TK_i$, $i = \overline{1...7}$.

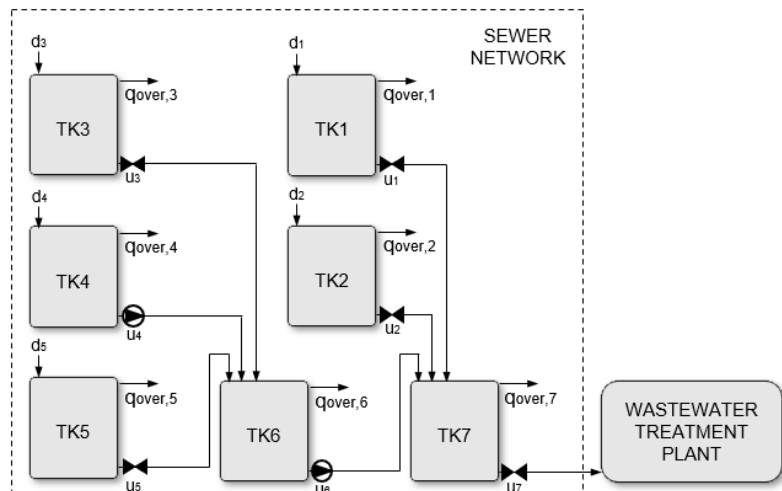

**Figure 1.** The structure of the sewer network.

It was considered that the 7 tanks collect wastewater and stormwater from 5 collection areas. It should also be mentioned that the first 4 tanks collect domestic wastewater and stormwater, while the 5th tank collects industrial wastewater. In Figure 1, the following notations are used:

- $d_i$, $i = \overline{1...5}$, is the inflow of tank $i$ from the corresponding collection area;
- $u_i$, $i = \overline{1...7}$, is the outflow of tank $i$;
- $q_{over,i}$, $i = \overline{1...7}$, represents the overflow of each tank.

Each collection area is characterized by its surface and the number of inhabitants living on that surface, as stated in Table 1.

**Table 1.** The characteristics of the collection areas.

| Collection Area Number | Collection Area Surface (ha) | The Population Served (Number of Inhabitants) |
| --- | --- | --- |
| 1 | 2500 | 75,000 |
| 2 | 5000 | 50,000 |
| 3 | 6000 | 40,000 |
| 4 | 7500 | 60,000 |
| 5 | 3500 | 25,000 |

Tanks 1–4 and tank 7 communicate via free fall of the polluted water, using a valve at the outlet to control the outflow, while the other 2 tanks (tanks 5 and 6) use pumps to drain the wastewater.

The volume of each tank, $V_i$, is computed considering a value of 0.05 m$^3$ for each inhabitant of the collection area served by tank $i$ if it directly serves a collecting area or, conversely, 0.05 m$^3$ for each of the average number of inhabitants served by the tanks whose outflow is part of the influent of tank $i$. For the tank that serves the area containing the industrial zone, an additional capacity was added with the same value as the average industrial wastewater production per day—$Q_{ind} = 2500$ m$^3$/day.

The maximum flow at the output of tank $i$, $U_{max,i}$, is computed as follows:

- For the tanks that use pumps to drain the wastewater, a constant value is considered representing the maximum flow of the pump;
- For the gravitational tanks, the maximum flow is considered to be 3 times higher than the average wastewater production of the population served by tank $i$, or by

the tanks whose outflow is part of tank $i$'s inflow—the average domestic water $Q_{pe} = 0.15 \text{ m}^3/(\text{day}\cdot\text{inhabitant})$.

The volumes and the maximum output flow of the 7 tanks are given in Table 2.

**Table 2.** The characteristics of the tanks of the sewer network.

| Tank Number | The Tank Volume (m³) | Maximum Flow at the Tank Output (m³/day) |
|:---:|:---:|:---:|
| 1 | 3750 | 33,750 |
| 2 | 2500 | 22,500 |
| 3 | 2000 | 18,000 |
| 4 | 3000 | 15,000 |
| 5 | 3750 | 18,750 |
| 6 | 2083 | 25,000 |
| 7 | 2500 | 112,500 |

The following assumptions were considered for the transport component of each collection area [10]:

1. A 15 min delay until the wastewater travels from one person to the tank corresponding to the collection area;
2. Twenty-five percent of TSS can accumulate in the piping, but no more than 10 kg/ha. The accumulated particulates will be flushed away when the flow increases over 8 times compared to the average domestic flow of the corresponding collection area (e.g., during pluviometric events).

### 2.2. The Influent of the Sewer Network

The influent of the sewer network comes from the 5 collection areas previously mentioned. It is defined on a time horizon of 28 days. Domestic wastewater and stormwater are collected from the first 4 areas, while industrial wastewater is collected from the 5th area, with loads specific to a medium-sized brewery industry [30]. Daily and yearly variations in the influent were also considered. Thus, all three components of the collected wastewater (domestic, industrial, and stormwater) can be modeled by time series based on previous information and/or on meteorological information. Figure 2 presents the influent generated for the 1st collection area, while Figure 3 shows the influent generated for the 5th collection area (the industrial part of the city).

Table 3 gives the average loads and the average flow values of the entire time horizon of 28 days (39,968 values for each tank), expressed in kg/m³. Table 4 shows the same average loads for all of the collection areas, expressed as concentrations.

**Table 3.** Influent medium loads and flow values.

| Area Number | CODsol (kg/day) | CODpart (kg/day) | NH$_4^+$ (kg/day) | PO$_4^{3-}$ (kg/day) | Flow (m³/day) |
|:---:|:---:|:---:|:---:|:---:|:---:|
| 1 | 720.71 | 4607.6 | 220.84 | 57.05 | 7786.2 |
| 2 | 487.97 | 3104.7 | 143.80 | 38.26 | 5283.0 |
| 3 | 421.18 | 2693.0 | 121.91 | 30.95 | 4400.5 |
| 4 | 589.17 | 3789.9 | 182.30 | 44.36 | 6369.4 |
| 5 | 2742.60 | 2051.3 | 132.67 | 61.22 | 4771.9 |

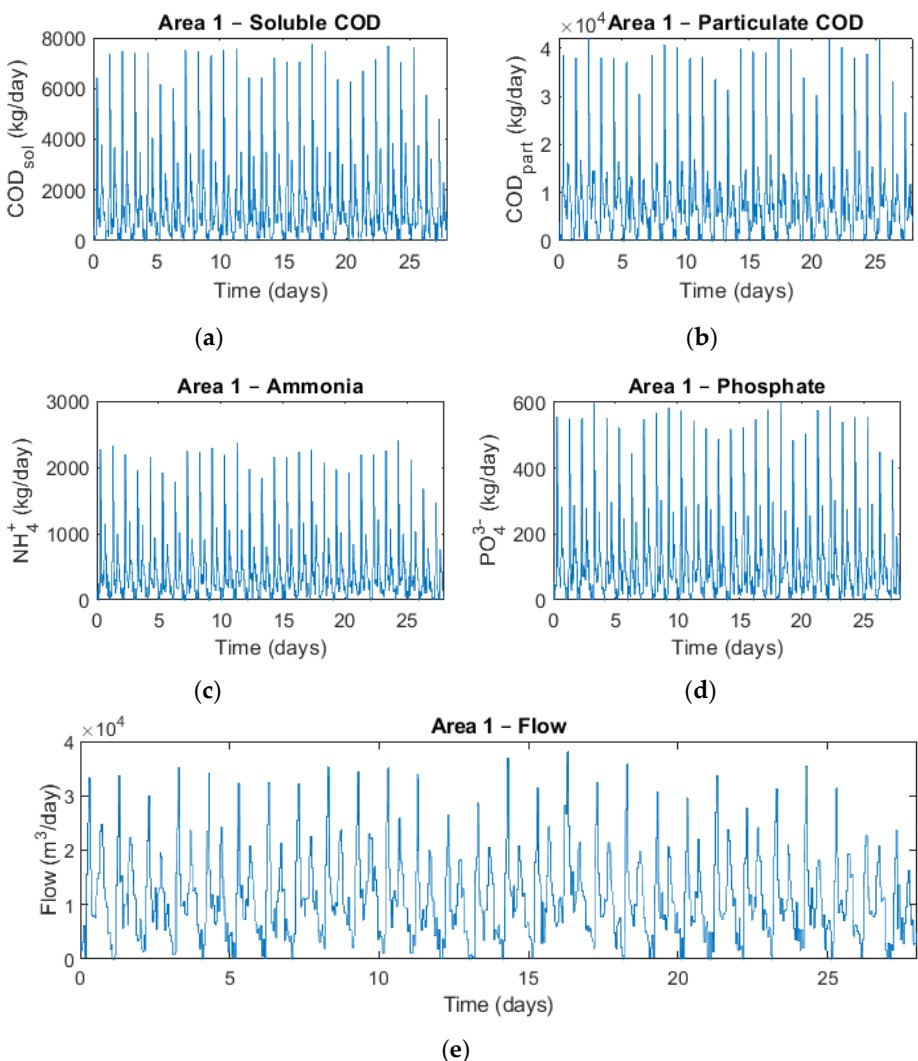

**Figure 2.** The influent corresponding to the 1st collection area: (**a**) soluble COD; (**b**) particulate COD; (**c**) ammonia; (**d**) phosphate; (**e**) flow of the 1st collection area.

**Table 4.** Influent medium loads as concentrations.

| Area Number | CODsol (g/m$^3$) | CODpart (g/m$^3$) | NH$_4^+$ (g/m$^3$) | PO$_4^{3-}$ (g/m$^3$) |
|:---:|:---:|:---:|:---:|:---:|
| 1 | 92.57 | 591.78 | 28.67 | 7.33 |
| 2 | 92.37 | 587.68 | 27.22 | 7.25 |
| 3 | 93.67 | 611.97 | 27.71 | 7.04 |
| 4 | 92.50 | 595.03 | 28.62 | 6.97 |
| 5 | 574.74 | 429.87 | 27.81 | 12.83 |

### 2.3. Performance Criteria of a Sewer Network

The efficiency of the operation of a sewerage network is given by 10 performance criteria implemented for BSMSewer [10], which are presented in Table 5.

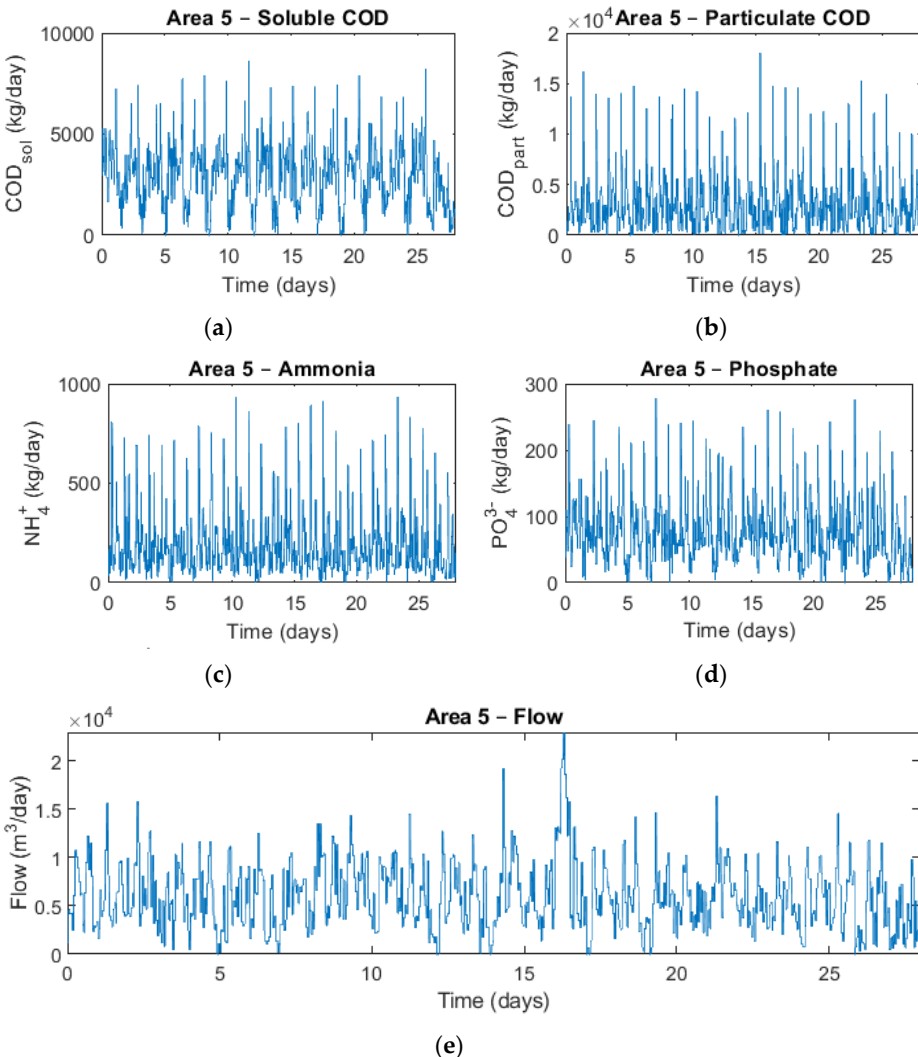

**Figure 3.** The influent corresponding to the 5th collection area: (**a**) soluble COD; (**b**) particulate COD; (**c**) ammonia; (**d**) phosphate; (**e**) flow of the 5th collection area.

### 2.4. Genetic Algorithm

The genetic algorithm is a metaheuristic algorithm commonly used to find solutions to the optimization problems, inspired by the biological natural selection process. It uses a population of chromosomes, consisting of genes, that is a genetic representation of the solutions to the optimization problem. It uses biologically inspired operators such as crossover, mutation, and selection to evolve from the initial population to a population containing solutions that minimize a fitness function associated with a chromosome. During each generation, the algorithm uses the selection operator to select a part of the existing population to breed a new generation. A new generation solution is obtained by applying the crossover function to its pair of parents, and by using the mutation operator to randomly change the values of some genes [31].

**Table 5.** Performance criteria.

| Criteria | Unit | Description |
|---|---|---|
| $N_{ovf}$ | 1/year | Frequency of overflow |
| $T_{ovf}$ | Days/year | Duration of overflow |
| $V_{ovf}$ | $m^3$/year | Volume of overflow |
| OQI | kg of polluting units/day | Overflow quality index |
| $C_{max,TSS}$ | $gCOD/m^3$ | Maximum concentration of TSS/hour |
| $C_{max,TKN}$ | $gN/m^3$ | Maximum concentration of TKN/hour |
| $C_{max,PO4}$ | $gP/m^3$ | Maximum concentration of $PO_4$/hour |
| $T_{exc,TSS}$ | Days/year | Annual duration of exceedances for TSS |
| $T_{exc,TKN}$ | Days/year | Annual duration of exceedances for TKN |
| $T_{exc,PO4}$ | Days/year | Annual duration of exceedances for $PO_4$ |

The genetic algorithm used to minimize the volume of overflow and the overflow quality index for a sewer network has the following characteristics:

- The chromosome number and structure of genes is chosen according to each control strategy approach, as can be seen in the following sections;
- The population size is chosen to consist of 20 individuals;
- The initial population is randomly generated with a uniform distribution;
- The crossover function that generates the offspring chromosome (*C*) from two parent chromosomes (*P1* and *P2*) generates a binary vector (*S*) with random values that have the same length as the chromosome. Considering this randomly generated binary vector, if *S(k)* = 0, then the gene *k* of the offspring chromosome inherits the value of gene *k* of the first parent, or if *S(k)* = 1, the gene *k* of the offspring chromosome inherits the gene *k* value of the second parent (Figure 4).
- The parent chromosomes are randomly chosen from the population, using for each candidate a probability directly proportional to its fitness function;
- To each gene of the parents, he mutation function adds a random value chosen from a Gaussian distribution with zero mean and standard deviation calculated for each gene as follows:

$$\sigma_k = \sigma_{k-1} \times \left(1 - \frac{k}{Gen}\right) \tag{1}$$

where

  - *k*—the current generation number;
  - *Gen*—the maximum number of generations;
  - $\sigma_k$—standard deviation of the Gaussian distribution at the generation *k* (at the first generation, $\sigma_1$ is the size of the gene range);

- The genetic algorithm stops when the maximum number of generations is reached (100 generations), or when the best fitness does not improve in the last 30 generations.

Remark: For each of the control strategies, the genetic algorithm was run twice, with two different fitness functions. The first time, it looked for the optimal controls to minimize the volume of overflow ($V_{ovf}$), while the second time it minimized the overflow quality index (OQI).

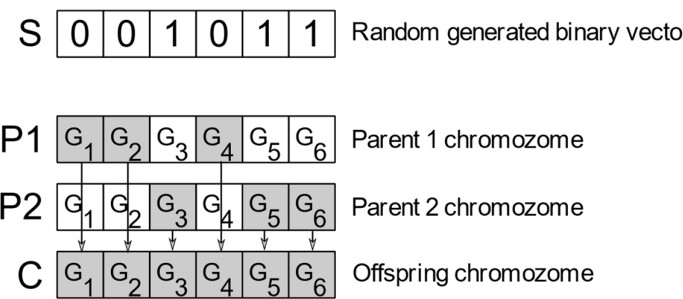

**Figure 4.** Genetic algorithm: crossover operator.

## 3. Simulation Results

The optimization of the operation of the sewer network is based on the prediction of the influent (as shown in Section 2.2) and on the control computation using genetic algorithms (Figure 5).

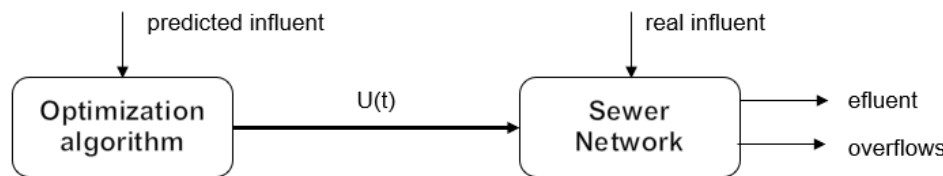

**Figure 5.** Optimal control of the sewer network.

Five strategies for optimizing the operating of the sewer network were approached. The analysis of the sewer network efficiency was focused mainly on 2 of the performance criteria—$V_{ovf}$ and OQI—of the 10 mentioned in Section 2.3.

### 3.1. Operation of the Sewer Network with Constant Controls at Maximum Values (100%)

This case was considered in [29]. No control system was taken into account; therefore, all valves at the tank outputs were considered to be fully opened, while all of the pumps were running at 100% as long as the wastewater level in the tank was over 0.5 m. The simulation results are presented in Table 6.

**Table 6.** Simulation results (no control is taken into account).

| Tank Number | $V_{ovf}$ (m³/year) | OQI |
|:---:|:---:|:---:|
| 1 | 0 | 0 |
| 2 | 22,729 | 192 |
| 3 | 50,202 | 743 |
| 4 | 55,134 | 1365 |
| 5 | 0 | 0 |
| 6 | 241,600 | 3830 |
| 7 | 0 | 0 |
| Global | 369,665 | 6131 |

Analyzing the sewer network structure from Figure 1 and the results from Table 6, it can be observed that minimizing the two performance criteria of the sewer network can be done only in relation with the output controls of tanks 3, 4, and 5. Setting the output control of the other tanks to a value smaller than 100% does not help in decreasing any of the performance criteria. The outflows of tanks 1, 2, and 6 represent the inflows for tank 7 (Figure 1) and, because tank 7 does not overflow when no control is used, setting the

output control of these tanks to a value smaller than 100% does not help in decreasing any of the performance criteria. The only effect of decreasing the controls of tanks 1, 2, and 6 is to increase the overflows of those tanks. The outflow of tank 7 is connected directly to the wastewater treatment plant input, so keeping this tank's output control set to 100% is the best solution in relation with the sewer network performance.

*3.2. Control Strategy #1—Constant Controls*

In this strategy, the controls at the outputs of tanks 3, 4, and 5 are maintained constant at certain values that are determined by the genetic algorithm in order to minimize the fitness function. The chromosome contains three genes (Figure 6):

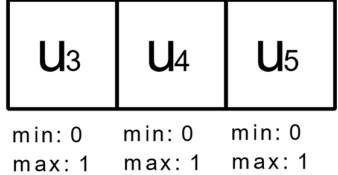

**Figure 6.** Chromosome structure in Strategy #1.

where $u_i \in [0,\ 1]$ is the percentage control of the valve/pump at the outlet of tank $i$.

3.2.1. Volume of Overflow Minimization (Strategy #1)

Characteristics: fitness function: $V_{ovf}$; the algorithm stopped after 69 generations.
In Figure 7, the best and the average fitness of each of the generations can be observed.

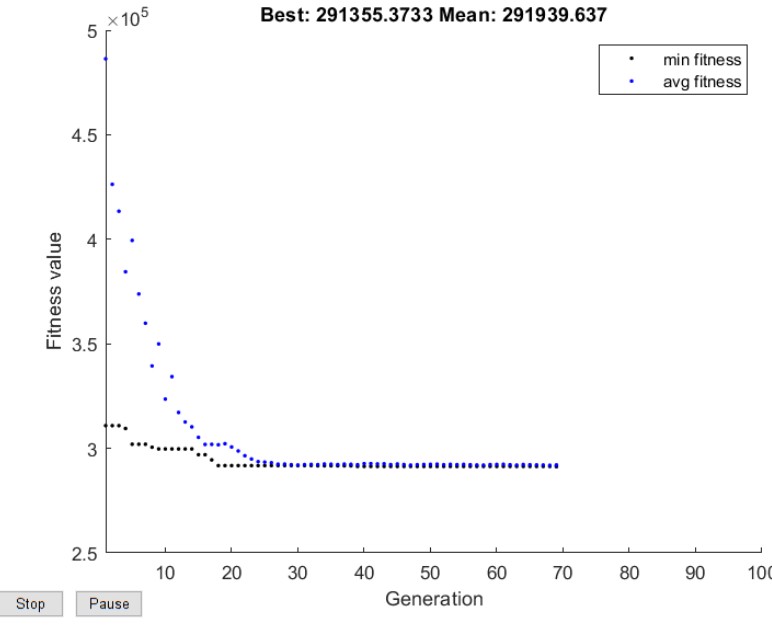

**Figure 7.** Best and average fitness evolution in control strategy #1 when minimizing $V_{ovf}$.

The following optimal controls were obtained:

$$\begin{bmatrix} u_3^* = 0.7 & u_4^* = 0.8743 & u_5^* = 0.6543 \end{bmatrix} \tag{2}$$

The simulation results are presented in Table 7.

**Table 7.** Simulation results when the volume of overflow is minimized (Strategy #1).

| Tank Number | $V_{ovf}^{*}$ (m³/year) | OQI |
|:---:|:---:|:---:|
| 1 | 0 | 0 |
| 2 | 22,750 | 192 |
| 3 | 63,727 | 886 |
| 4 | 61,099 | 1376 |
| 5 | 12,924 | 451 |
| 6 | 130,855 | 2223 |
| 7 | 0 | 0 |
| Global | 291,355 | 5128 |

\* Optimal value of $V_{ovf}$.

### 3.2.2. Overflow Quality Index Minimization (Strategy #1)

Characteristics: fitness function: OQI; the algorithm stopped after 91 generations. Figure 8 presents the best and the average fitness evolution during the generations.

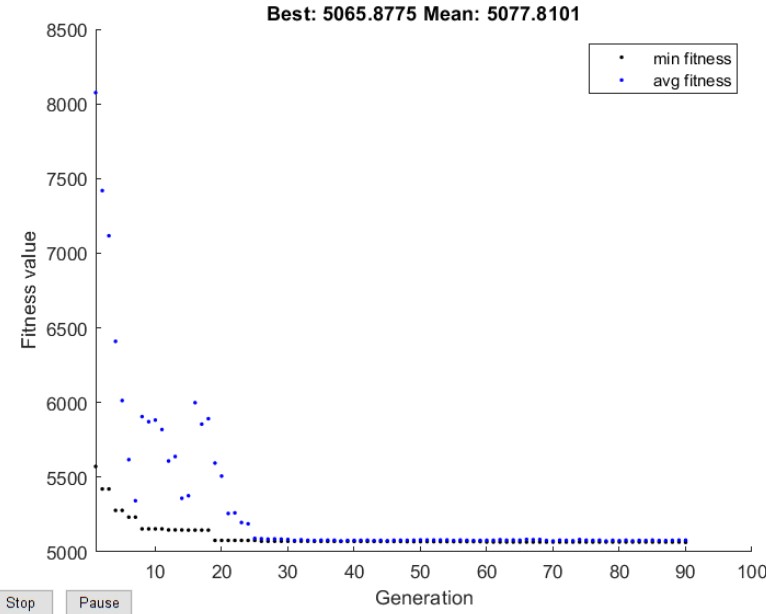

**Figure 8.** Best and average fitness evolution in control strategy #1 when minimizing OQI.

The values of the optimal controls are as follows:

$$\begin{bmatrix} u_3^* = 0.7062 & u_4^* = 0.6872 & u_5^* = 0.6742 \end{bmatrix} \tag{3}$$

The simulation results are presented in Table 8.

### 3.3. Control Strategy #2

The outputs of tanks 3, 4, and 5 are considered to vary in time, depending on the liquid level in the tank, as follows: if the liquid level is below a certain value $h_{i,LIM}$, the valve/pump control will be $u_i$, while if the level is above $h_{i,LIM}$, then the control will be considered 1 (100%). This way, when the liquid level is over a certain limit, the output control is kept to 100%, so the overflow of the current tank will be as low as possible. In order to prevent the quick change of the control when the tank level is around $h_{i,LIM}$, a hysteresis was considered, as presented in Figure 9.

**Table 8.** Simulation results when the overflow quality index is minimized (strategy #1).

| Tank Number | $V_{ovf}$ (m³/year) | OQI * |
|:---:|:---:|:---:|
| 1 | 0 | 0 |
| 2 | 22,740 | 193 |
| 3 | 63,189 | 852 |
| 4 | 106,189 | 1789 |
| 5 | 11,876 | 422 |
| 6 | 103,051 | 1811 |
| 7 | 0 | 0 |
| Global | 307,044 | 5066 |

* Optimal value of OQI.

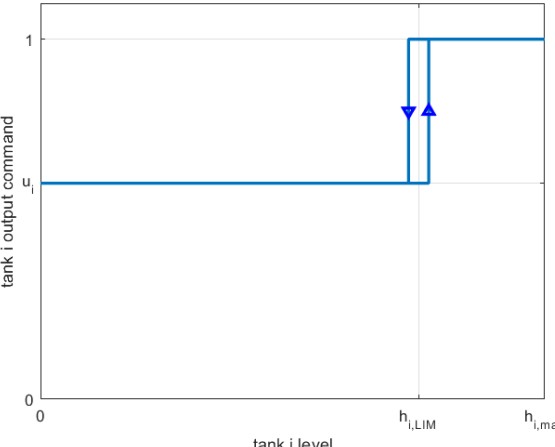

**Figure 9.** Strategy #2 controls depending on the liquid level in the tanks.

For each of the three tanks, $h_{i,LIM}$ and $u_i$, $i = 3 \ldots 5$, are determined by the optimization algorithm in order to minimize the fitness function. The chromosomal structure is presented in Figure 10.

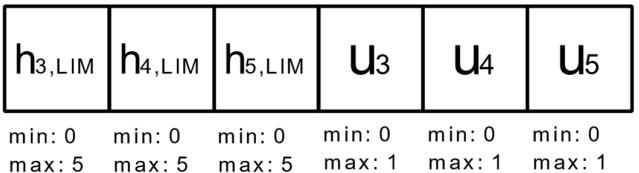

**Figure 10.** Chromosomal structure in strategy #2.

### 3.3.1. Volume of Overflow Minimization (Strategy #2)

Characteristics: fitness function: $V_{ovf}$; the algorithm stopped after 93 generations.
In Figure 11 the best and the average fitness of each of the generations can be observed. The following optimal solutions were obtained:

$$\begin{bmatrix} h^*_{3,LIM} = 4.9192 & h^*_{4,LIM} = 0.2053 & h^*_{5,LIM} = 4.9896 \\ u^*_3 = 0.7716 & u^*_4 = 0.7983 & u^*_5 = 0.6096 \end{bmatrix} \quad (4)$$

The simulation results are presented in Table 9.

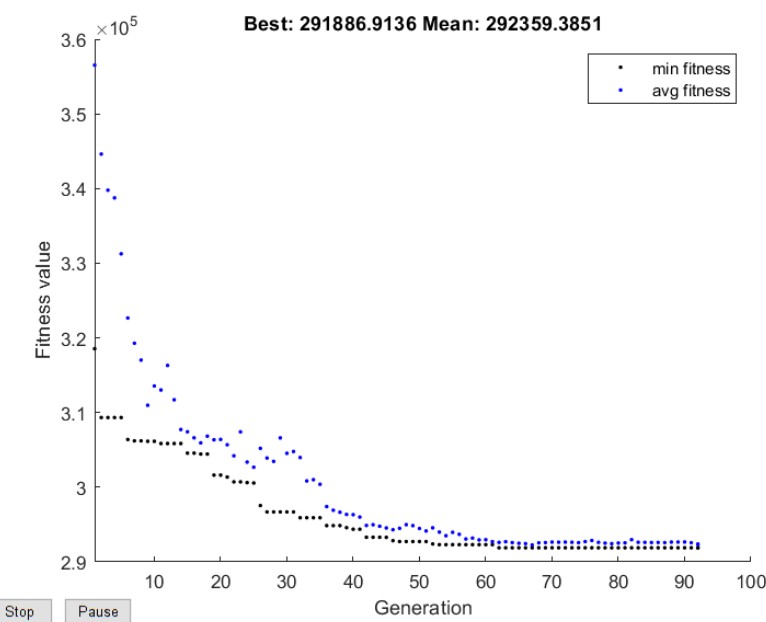

**Figure 11.** Best and average fitness evolution in control strategy #2 when minimizing $V_{ovf}$.

**Table 9.** Simulation results when the volume of overflow is minimized (strategy #2).

| Tank Number | $V^*_{ovf}$ (m³/year) | OQI |
| :---: | :---: | :---: |
| 1 | 0 | 0 |
| 2 | 22,730 | 192 |
| 3 | 57,059 | 784 |
| 4 | 55,156 | 1365 |
| 5 | 15,512 | 508 |
| 6 | 141,430 | 2307 |
| 7 | 0 | 0 |
| Global | 291,887 | 5157 |

* Optimal value of $V_{ovf}$.

### 3.3.2. Overflow Quality Index Minimization (Strategy #2)

Characteristics: fitness function: OQI; the algorithm stopped after 100 generations. Figure 12 shows the best and the average fitness of each of the generations. The following optimal controls were obtained:

$$
\begin{bmatrix}
h^*_{3,LIM} = 4.8716 & h^*_{4,LIM} = 4.7138 & h^*_{5,LIM} = 3.3904 \\
u^*_3 = 0.6206 & u^*_4 = 0.7312 & u^*_5 = 0.9158
\end{bmatrix}
\tag{5}
$$

### 3.4. Control Strategy #3

The controls for the outputs of tanks 3, 4, and 5 are considered to vary in time, depending on the liquid level in the tank, as follows: if the liquid level in the tank is below a certain level $h_{i,LIM}$, the valve/pump control will be $u_{i,B}$, while if the level is above $h_{i,LIM}$, then the control will be considered $u_{i,A}$. No inequality constraints for $u_{i,A}$ and $u_{i,B}$ were considered. In this way, when the liquid level is approaching the limit, the new control value can be higher than the old one, in order to make the overflow of the current tank smaller, or it can be lower in order to help the tank whose inflow is the outflow of the current tank to overflow less. To prevent the control quickly changing when the tank level is around $h_{i,LIM}$, a hysteresis was considered around this level, as presented in Figure 13.

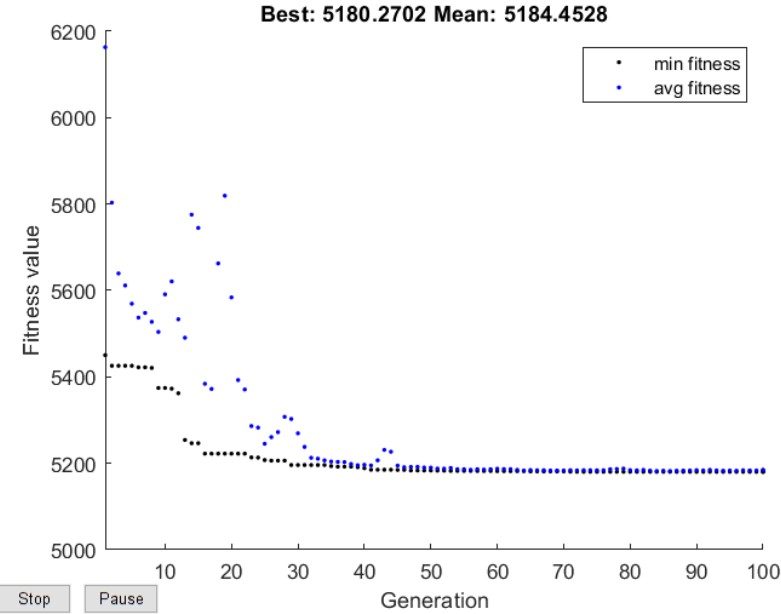

**Figure 12.** Best and average fitness evolution in control strategy #2 when minimizing OQI.

**Table 10.** Simulation results when the overflow quality index is minimized (strategy #2).

| Tank Number | $V_{ovf}$ (m³/year) | OQI [*] |
|:---:|:---:|:---:|
| 1 | 0 | 0 |
| 2 | 22,728 | 192 |
| 3 | 66,739 | 878 |
| 4 | 76,156 | 1488 |
| 5 | 535 | 15 |
| 6 | 156,137 | 2607 |
| 7 | 0 | 0 |
| Global | 322,295 | 5180 |

[*] Optimal value of OQI.

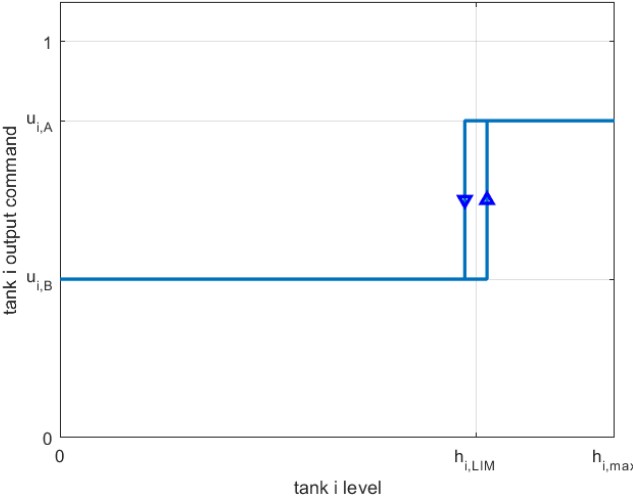

**Figure 13.** Strategy #3 controls depending on the liquid level in the tanks.

For each of the three tanks, $h_{i,LIM}$, $u_{i,A}$, and $u_{i,B}$, $i = 3 \ldots 5$ are determined by the optimization algorithm to minimize the fitness function. In this case, the chromosomal structure is presented in Figure 14.

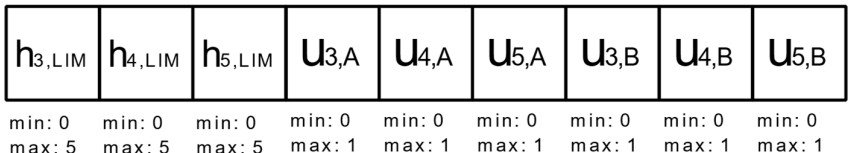

**Figure 14.** Strategy #3 chromosomal structure.

### 3.4.1. Volume of Overflow Minimization (Strategy #3)

Characteristics: fitness function: $V_{ovf}$; the algorithm stopped after 100 generations. Figure 15 shows the best and the average fitness of each of the generations.

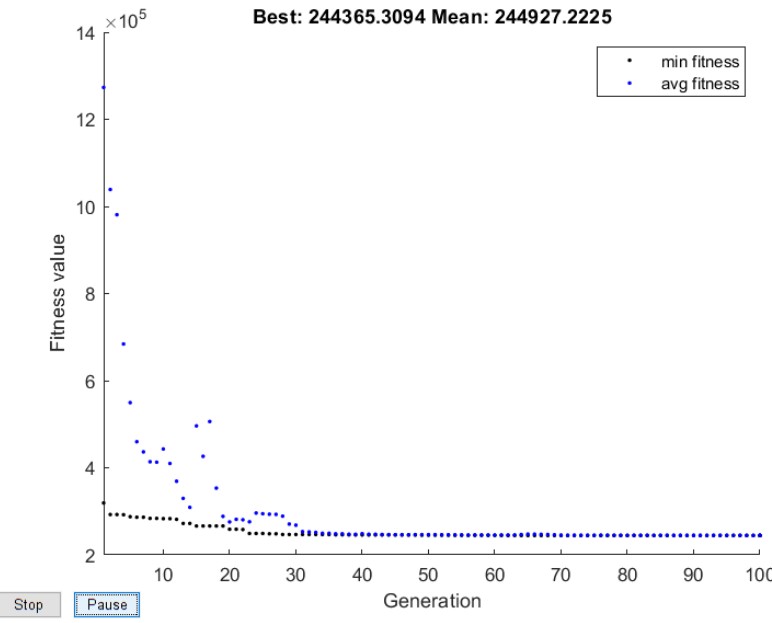

**Figure 15.** Best and average fitness evolution in control strategy #3 when minimizing $V_{ovf}$.

The following optimal solutions were obtained:

$$
\begin{bmatrix}
h^*_{3,LIM} = 3.507 & h^*_{4,LIM} = 4.3294 & h^*_{4,LIM} = 2.0954 \\
u^*_{3,A} = 0.6997 & u^*_{4,A} = 0.6910 & u^*_{5,A} = 0.51 \\
u^*_{3,B} = 0.8423 & u^*_{4,B} = 0.8927 & u^*_{5,B} = 0.9836
\end{bmatrix}
\tag{6}
$$

The simulation results are presented in Table 11.

### 3.4.2. Overflow Quality Index Minimization

Characteristics: fitness function: OQI; the algorithm stopped after 100 generations. Figure 16 shows the best and the average fitness of each of the generations. The following optimal solutions were obtained:

$$
\begin{bmatrix}
h^*_{3,LIM} = 3.3937 & h^*_{4,LIM} = 0.8911 & h^*_{5,LIM} = 3.9285 \\
u^*_{3,A} = 0.5510 & u^*_{4,A} = 0.8598 & u^*_{5,A} = 0.6190 \\
u^*_{3,B} = 0.9157 & u^*_{4,B} = 0.8414 & u^*_{5,B} = 0.9961
\end{bmatrix}
\tag{7}
$$

The simulation results are presented in Table 12.

**Table 11.** Simulation results when the volume of overflow is minimized (strategy #3).

| Tank Number | $V^*_{ovf}$ (m$^3$/year) | OQI |
|:---:|:---:|:---:|
| 1 | 0 | 0 |
| 2 | 22,759 | 193 |
| 3 | 90,614 | 1197 |
| 4 | 85,118 | 1819 |
| 5 | 21,207 | 727 |
| 6 | 24,667 | 318 |
| 7 | 0 | 0 |
| Global | 244,365 | 4254 |

* Optimal value of $V_{ovf}$.

**Figure 16.** Best and average fitness evolution in control strategy #3 when minimizing OQI.

**Table 12.** Simulation results when the overflow quality index is minimized (strategy #3).

| Tank Number | $V_{ovf}$ (m$^3$/year) | OQI$^*$ |
|:---:|:---:|:---:|
| 1 | 0 | 0 |
| 2 | 22,772 | 193 |
| 3 | 156,472 | 1982 |
| 4 | 69,088 | 1526 |
| 5 | 9111 | 299 |
| 6 | 17,609 | 280 |
| 7 | 0 | 0 |
| Global | 275,052 | 4280 |

* Optimal value of OQI.

### 3.5. Control Strategy #4

The controls for the outputs of tanks 3, 4, and 5 are considered to vary in time, depending on the liquid level in the tank, as follows: if the liquid level in the tank is below a certain level $h_{i,LOW}$, the valve/pump control will be $u_{i,B}$; if the level is above $h_{i,HIGH}$, then

the control will be $u_{i,A}$, and if the level is between $h_{i,LOW}$ and $h_{i,HIGH}$, the control will be scaled accordingly between $u_{i,B}$ and $u_{i,A}$ (Figure 17).

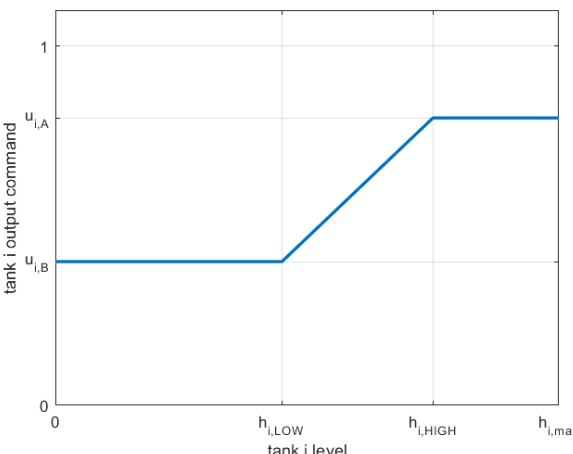

**Figure 17.** Strategy #4 control depending on the liquid level in the tank.

For each of the three tanks, $h_{i,LOW}$, $h_{i,HIGH}$, $u_{i,A}$, and $u_{i,B}$, $i = 3 \ldots 5$ are determined by the optimization algorithm to minimize the fitness function. The two limits, $h_{i,LOW}$ and $h_{i,HIGH}$, must always respect the constraint $h_{i,LOW} < h_{i,HIGH}$. This requires changing some characteristics of the genetic algorithm (e.g., initial population, crossover function, mutation function) to make sure that all solutions are valid. To avoid this, it must be considered that $h_{i,LOW} = \min(h_{i,1}, h_{i,2})$ and $h_{i,HIGH} = \max(h_{i,1}, h_{i,2})$, where $h_{i,1}$ and $h_{i,2}$ are provided by the genetic algorithm. Therefore, the chromosome contains 12 genes, as presented in Figure 18.

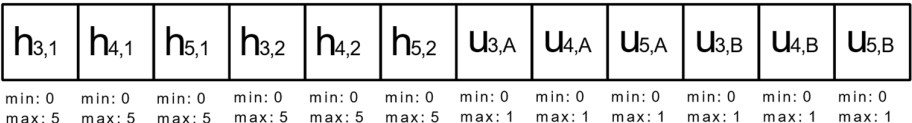

**Figure 18.** Strategy #4 chromosomal structure.

### 3.5.1. Volume of Overflow Minimization

Characteristics: fitness function: $V_{ovf}$; the algorithm stopped after 100 generations. Figure 19 presents the best and the average fitness of each of the generations. The following optimal solution is obtained:

$$
\begin{bmatrix}
h_{3,LOW}^* = 1.5247 & h_{4,LOW}^* = 2.6187 & h_{5,LOW}^* = 0.0654 \\
h_{3,HIGH}^* = 4.9168 & h_{4,HIGH}^* = 2.8814 & h_{5,HIGH}^* = 1.4234 \\
u_{3,A}^* = 0.6678 & u_{4,A}^* = 0.6564 & u_{5,A}^* = 0.6403 \\
u_{3,B}^* = 0.9829 & u_{4,B}^* = 0.9976 & u_{5,B}^* = 0.8426
\end{bmatrix}
\tag{8}
$$

The simulation results are presented in Table 13.

### 3.5.2. Overflow Quality Index Minimization

Characteristics: fitness function: OQI; the algorithm stopped after 100 generations. Figure 20 shows the best and the average fitness of each of the generations.

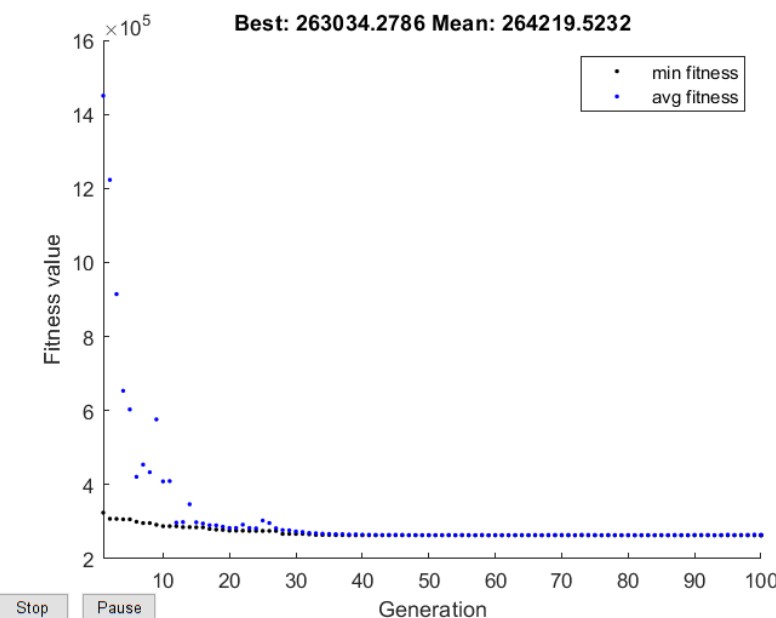

**Figure 19.** Best and average fitness evolution in control strategy #4 when minimizing $V_{ovf}$.

**Table 13.** Simulation results when the volume of overflow is minimized (strategy #4).

| Tank Number | $V^*_{ovf}$ (m$^3$/year) | OQI |
|:---:|:---:|:---:|
| 1 | 0 | 0 |
| 2 | 22,743 | 193 |
| 3 | 87,101 | 1168 |
| 4 | 86,162 | 1882 |
| 5 | 26,715 | 883 |
| 6 | 40,314 | 503 |
| 7 | 0 | 0 |
| Global | 263,034 | 4629 |

* Optimal value of $V_{ovf}$.

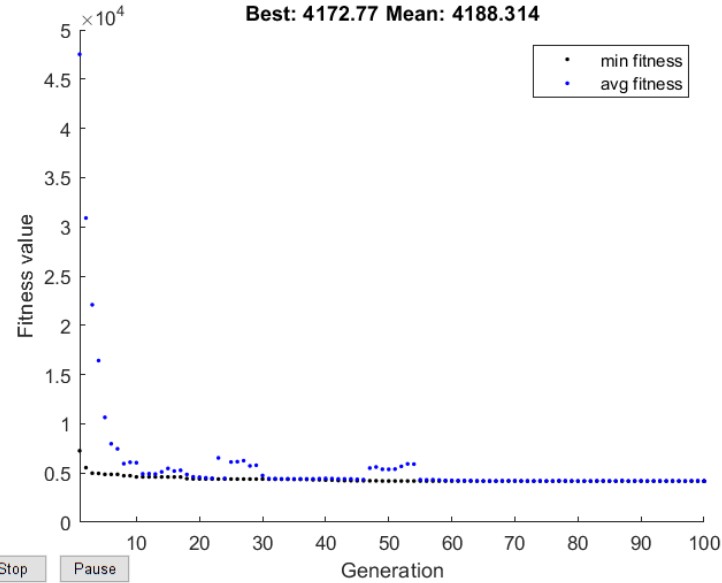

**Figure 20.** Best and average fitness evolution in control strategy #4 when minimizing OQI.

The following optimal solutions were obtained:

$$
\begin{bmatrix}
h^*_{3,LOW} = 2.265 & h^*_{4,LOW} = 0.3715 & h^*_{5,LOW} = 1.5886 \\
h^*_{3,HIGH} = 3.672 & h^*_{4,HIGH} = 3.6782 & h^*_{5,HIGH} = 2.53 \\
u^*_{3,A} = 0.622 & u^*_{4,A} = 0.8374 & u^*_{5,A} = 0.6492 \\
u^*_{3,B} = 0.8871 & u^*_{4,B} = 0.9323 & u^*_{5,B} = 0.9937
\end{bmatrix}
\tag{9}
$$

The simulation results are presented in Table 14.

**Table 14.** Simulation results when the overflow quality index is minimized (strategy #4).

| Tank Number | $V_{ovf}$ (m³/year) | OQI * |
|:---:|:---:|:---:|
| 1 | 0 | 0 |
| 2 | 22,759 | 193 |
| 3 | 126,250 | 1609 |
| 4 | 69,097 | 1547 |
| 5 | 15,677 | 539 |
| 6 | 18,166 | 258 |
| 7 | 0 | 0 |
| Global | 251,949 | 4173 |

* Optimal value of OQI.

### 3.6. Control Strategy #5

This strategy is similar to strategy #3 (Figure 13), but this time, it is considered that $h_{i,LIM}$ is equal to 3.5 m. This way, the number of genes of the chromosome is smaller, decreasing the number of dimensions of the solution space. This should improve the convergence of the algorithm and decrease the probability of getting stuck at a local minimum.

For each of the three tanks, $u_{i,A}$ and $u_{i,B}$, $i = 3 \ldots 5$, are determined by the optimization algorithm to minimize the fitness function. In this case, the structure of the chromosome is as presented in Figure 21.

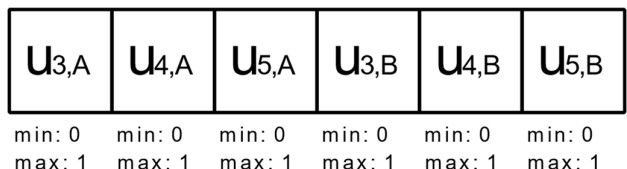

**Figure 21.** Strategy #5 chromosomal structure.

#### 3.6.1. Volume of Overflow Minimization

Characteristics: fitness function: $V_{ovf}$; the algorithm stopped after 100 generations. Figure 22 presents the best and the average fitness of each of the generations. The following optimal solutions were obtained:

$$
\begin{bmatrix}
u^*_{3,A} = 0.6085 & u^*_{4,A} = 0.8033 & u^*_{5,A} = 0.5609 \\
u^*_{3,B} = 0.9462 & u^*_{4,B} = 0.9008 & u^*_{5,B} = 0.7024
\end{bmatrix}
\tag{10}
$$

The simulation results are presented in Table 15.

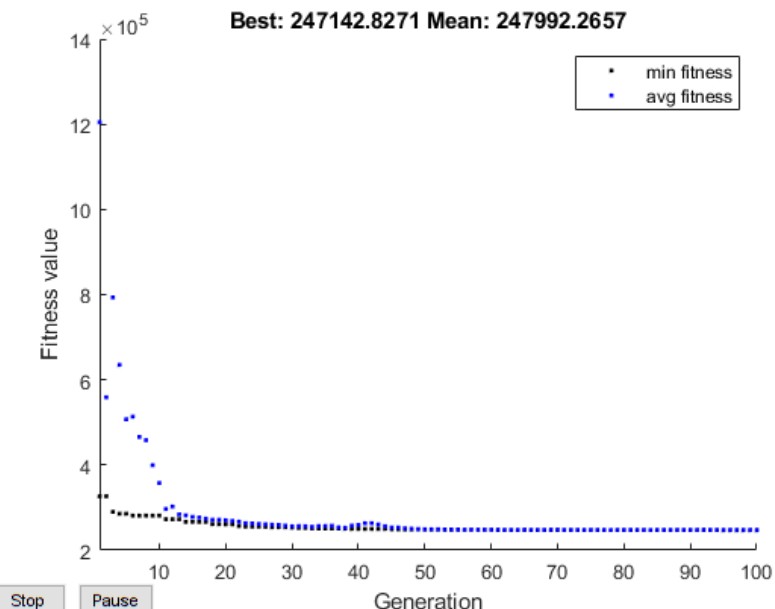

**Figure 22.** Best and average fitness evolution in control strategy #5 when minimizing $V_{ovf}$.

**Table 15.** Simulation results when the volume of overflow is minimized (strategy #5).

| Tank Number | $V^*_{ovf}$ (m³/year) | OQI |
| --- | --- | --- |
| 1 | 0 | 0 |
| 2 | 22,759 | 193 |
| 3 | 113,086 | 1489 |
| 4 | 72,589 | 1610 |
| 5 | 27,726 | 924 |
| 6 | 10,983 | 157 |
| 7 | 0 | 0 |
| Global | 247,143 | 4373 |

* Optimal value of $V_{ovf}$.

### 3.6.2. Overflow Quality Index Minimization

Characteristics: fitness function: OQI; the algorithm stopped after 100 generations. Figure 23 shows the best and the average fitness of each of the generations. The following optimal solutions were obtained:

$$\begin{bmatrix} u^*_{3,A} = 0.6626 & u^*_{4,A} = 0.4811 & u^*_{5,A} = 0.9417 \\ u^*_{3,B} = 0.9060 & u^*_{4,B} = 0.9955 & u^*_{5,B} = 0.9013 \end{bmatrix} \tag{11}$$

The simulation results are presented in Table 16.

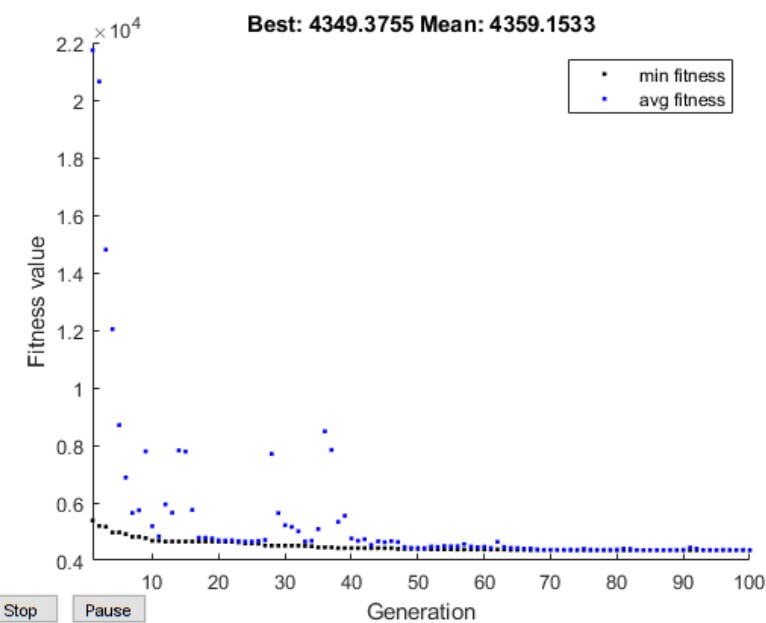

**Figure 23.** Best and average fitness evolution in control strategy #5 when minimizing OQI.

**Table 16.** Simulation results when the overflow quality index is minimized (strategy #5).

| Tank Number | $V_{ovf}$ (m³/year) | OQI * |
|---|---|---|
| 1 | 0 | 0 |
| 2 | 22,741 | 193 |
| 3 | 96,103 | 1275 |
| 4 | 105,689 | 2218 |
| 5 | 2844 | 89 |
| 6 | 45,448 | 574 |
| 7 | 0 | 0 |
| Global | 272,826 | 4349 |

* Optimal value of OQI.

## 4. Discussion

The study carried out in this paper presents a series of results regarding the optimization of the volume of overflow and the overflow quality index—results obtained on the model of a sewer network corresponding to a city in eastern Romania, with a population of 250,000 inhabitants. The method of genetic algorithms was used for optimization. Simple controls that required the computation of 3–12 parameters were generated.

Figure 24 presents the evolutions of the fitness functions over generations for all five strategies in the two cases considered in the analysis: (a) when minimizing $V_{ovf}$, and (b) when minimizing OQI.

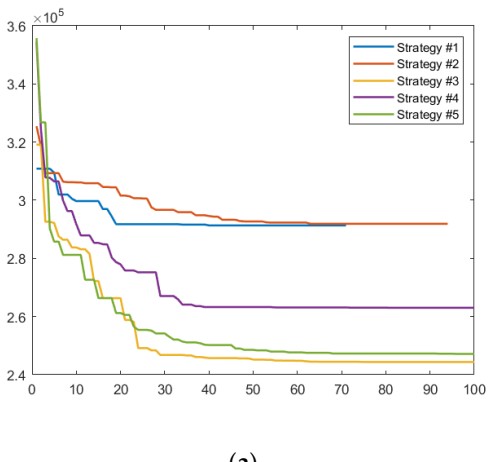
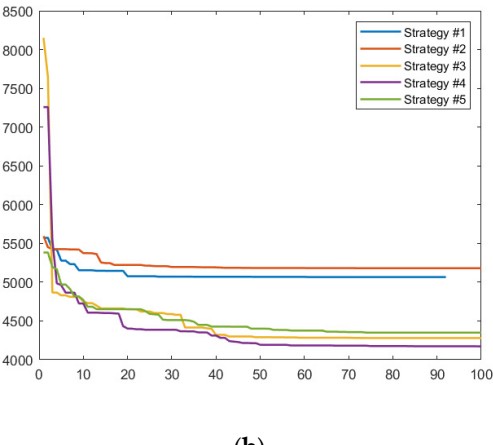

(**a**)            (**b**)

**Figure 24.** Fitness function evolutions for all strategies: (a) when minimizing $V_{ovf}$; (b) when minimizing OQI.

A synthesis of the simulation results obtained with the five optimal control strategies, in relation to the minimization of both the volume of overflow and the overflow quality index, is given in Table 17.

**Table 17.** Simulation results for solutions of the five strategies with both fitness functions.

| | $V^{*}_{ovf}$ (m$^3$/year) | Percentage of "No Control" | OQI $^*$ | Percentage of "No Control" |
|---|---|---|---|---|
| No control | 369,655 | | 6131 | |
| Strategy #1 (min $V_{ovf}$) | 291,355 | −21.18% | 5128 | −16.36% |
| Strategy #1 (min OQI) | 307,044 | −16.94% | 5066 | −17.37% |
| Strategy #2 (min $V_{ovf}$) | 291,877 | −21.04% | 5157 | −15.89% |
| Strategy #2 (min OQI) | 322,295 | −12.81% | 5180 | −15.51% |
| Strategy #3 (min $V_{ovf}$) | 244,365 | −33.89% | 4245 | −30.61% |
| Strategy #3 (min OQI) | 275,052 | −25.59% | 4280 | −30.19% |
| Strategy #4 (min $V_{ovf}$) | 263,034 | −28.84% | 4629 | −24.50% |
| Strategy #4 (min OQI) | 251,949 | −31.84% | 4173 | −31.94% |
| Strategy #5 (min $V_{ovf}$) | 247,143 | −33.14% | 4373 | −28.67% |
| Strategy #5 (min OQI) | 272,826 | −26.19% | 4349 | −29.06% |

$^*$ Optimal values of $V_{ovf}$ and OQI.

The results are analyzed by comparison with the case when the controls have maximum values (the valves are fully opened and the pumps provide maximum flows). In the third and fifth columns (Percentage of "No control"), the differences can be found between the values of the performance criteria computed when the optimization strategies are applied and the aforementioned situation. For example, −21.18% (column #3) means that strategy #1 ensures a 21.18% lower volume of overflow than in the case of "No control", which means that strategy #1 is more efficient in relation to this performance criterion.

The results from Table 17 show that in all of the optimization strategies applied in the conditions of an influent that contains the three components of wastewater (domestic, industrial, and stormwater), significant improvements in the performance criteria were observed. Of the five optimization strategies, strategy #3 ensures the most efficient operation in relation to the volume of overflow, while strategy #4 ensures the best efficiency in relation to the overflow quality index, as highlighted in Table 17 in yellow. As the values of the overflow quality index are quite close in strategies #3 and #4 (−30.19% compared to

−31.94%), it can be stated that strategy #3 can be considered the best. In fact, the evolutions of the fitness functions (Figure 24a—orange curve and Figure 24b—magenta curve) show the same thing—they converge to the minimum values.

Statistical parameters for the strategy that gave the best results (strategy #3, when minimizing $V_{ovf}$) were computed. The genetic algorithm was run 18 times, and the results shown in Table 18 were obtained.

**Table 18.** 18 results of strategy #3 when minimizing $V_{ovf}$.

| Run Number | $V_{ovf}^*$ (m³/year) | Percentage of "No Control" | OQI | Percentage of "No Control" |
|---|---|---|---|---|
| 1 | 244,365 | −33.89% | 4245 | −30.61% |
| 2 | 261,818 | −29.17% | 4628 | −24.51% |
| 3 | 291,646 | −21.10% | 5003 | −18.39% |
| 4 | 245,079 | −33.70% | 4322 | −29.50% |
| 5 | 279,821 | −24.30% | 4932 | −19.55% |
| 6 | 269,879 | −26.99% | 4824 | −21.31% |
| 7 | 277,738 | −24.86% | 4858 | −20.76% |
| 8 | 245,616 | −33.55% | 4265 | −30.43% |
| 9 | 256,327 | −30.65% | 4476 | −26.99% |
| 10 | 241,795 | −34.58% | 4339 | −29.22% |
| 11 | 254,391 | −31.18% | 4554 | −25.72% |
| 12 | 265,862 | −28.07% | 4621 | −24.62% |
| 13 | 265,846 | −28.08% | 4756 | −22.28% |
| 14 | 253,098 | −31.53% | 4375 | −28.93% |
| 15 | 258,943 | −29.95% | 4429 | −27.76% |
| 16 | 254,574 | −31.13% | 4465 | −27.17% |
| 17 | 251,778 | −31.88% | 4405 | −28.15% |
| 18 | 248,581 | −32.75% | 4375 | −28.64% |

* Optimal value of $V_{ovf}$.

Table 19 shows the values of the statistical parameters in relation to the two criteria ($V_{ovf}$ and OQI):

**Table 19.** Statistical parameters.

| | Maximum Value | Minimum Value | Mean Value | Standard Deviation |
|---|---|---|---|---|
| $V_{ovf}^*$ (m³/year) | 291,646 (−21.10%) | 241,795 (−34.58%) | 259,286.5 (−29.85%) | 13,710.8951 |
| OQI | 5003 (−18.39%) | 4245 (−30.61%) | 4548.4444 (−25.81%) | 237.0131 |

* Optimal value of $V_{ovf}$.

Low values of the standard deviation were obtained in relation to both $V_{ovf}$ and OQI, showing that the values of the two performance criteria are quite well grouped around the average values. This proves that the algorithm is robust and reliable.

The optimization strategies approached in this paper assume the following: optimal controls are computed and applied in the process for the time horizon considered; during this period, a new set of controls is determined, which will be applied in the next time horizon, and so on. Compared to the present paper, in [32], the use of fuzzy-logic-based

structures for the control of storage tank capacities in order to reduce the volume of overflow is proposed. Although the results obtained by the authors are good, this method does not guarantee optimality.

Additionally, [18] deals with a similar problem following the control of a combined sewer network by two methods: nonlinear optimal control and multivariable feedback control. The case study consisted of a sewer network of lower capacity than the one used in this paper. In the case of the first method, the authors' attention was focused on two objectives: (1) increasing the efficiency of the sewer network operating only in terms of the volume of overflow—as opposed to the present paper, which also considered OQI—and (2) increasing the computational efficiency of the optimization algorithm. A qualitative analysis of the results obtained in the present paper and in [18] shows that they are similar, taking into account the differences in structure and scale between the two sewer networks.

## 5. Conclusions

The present paper shows that sewer networks are very well suited to the use of optimization algorithms in order to increase their operational performance. The main idea of this paper was to determine the optimal set of controls to ensure minimal values of two of the performance criteria proposed for the BSMSewer simulation environment (volume of overflow, and overflow quality index). Five optimization strategies based on genetic algorithms were proposed and tested on a varied influent containing three components: domestic wastewater, stormwater, and industrial wastewater.

In essence, our method has the advantage of simplicity in application. The accuracy of the results depends directly on the influent prediction over the chosen time horizon. In principle, the prediction horizon can be reduced so that the values of the influent have a higher degree of confidence. Through an analysis of meteorological observations, an acceptable compromise can be reached regarding the choice of the prediction horizon between the degree of confidence of the meteorological observations and the computing time. Furthermore, this paper shows that the results are better as the number of control parameters (chromosome structure) is increased (strategies #3 and #4 have 9 and 12 parameters, respectively, compared to strategy #1 (3 parameters), strategy #2 (6 parameters), and strategy #5 (6 parameters)). The price paid is the significant increase in the running time in the case of strategies #3 and #4, with 12–16 h optimization sequences, using a computer with the following characteristics: Intel Core i5-10500T CPU @ 2.30 GHz with 8 GB of RAM.

In subsequent research, the authors aim to approach other control (optimization) strategies for sewer networks: optimization of sewer networks using aggregate criteria (e.g., the two criteria considered in the paper aggregated in a single criterion), real-time control of sewer networks, and the use and testing of other optimization algorithms to solve the optimization problem, including the wastewater treatment plant in the optimization procedure.

**Author Contributions:** Conceptualization, I.V., S.C. and R.V.; software, I.V. and L.L.; methodology, M.B.; writing—review and editing, I.V. and S.C.; validation, I.V. and M.B. All authors have read and agreed to the published version of the manuscript.

**Funding:** This research was supported by the project "DINAMIC" financed by the Romanian Ministry of Research and Innovation, Contract no. 12PFE/2021. The research was also supported by the Spanish Government through the MICINN project PID2019-105434RB-C33.

**Data Availability Statement:** Not applicable.

**Conflicts of Interest:** The authors declare no conflict of interest.

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
