# Peer review of "Optimal Control Strategy of a Sewer Network"

_water, doi:10.3390/w14071062_

Round 1

Reviewer 1 Report

The manuscript „Optimal control strategy of a sewer network” has in attention the modelling of the sewer network intended for 250000 inhabitants.

The authors are kindly requested to consider the following recommendations:

  • The introduction should refer to the actual state-of-the-art and include more bibliographic references;
  • Row 127 - the authors should correctly insert the reference;
  • In the case, that Figure 1 is from reference 21, reprint permission is required;
  • For the values in Tables 3 and 4, please show how many values you considered for calculations;
  • The text in Figure 4 should be written with bigger characters;
  • standard deviations should be inserted for the results;
  • Carefully check all the abbreviations and insert them where they are firstly appearing (for example discharge quality index 
    (OQI) and Overflow Quality Index (OQI), there is no difference?);
  • The section Discussion should be extended and the results compared to other previous published literature in the research area;
  • It is commendable to have Conclusions and Future perspectives.

Reviewer 2 Report

I have gone through carefully the content and would like to inform that the paper is easy to understand but some chapters should be improved. The conclusions are missed and some items need to be completed. Some specific comments are as follows:

  • Introduction and Literature review: The authors analyzed the progress and problem of previous researches, however please provide more examples (papers). Additionally, please emphasize on works recently done (not older than 5-6 years).
  • Materials and methods: This chapter should be improved. L217. “Error..”? It should be ‘kg’, not “Kg”. Table 5 should be improved (especially the units: m3)
  • Results: Figure 7, 13, 16, 18 are unreadable.
  • Discussion/ conclusions- it seems that chapter 4 should be called “conclusions” so the discussion is missing??? References should be added in this chapter definitely. It should be specified.
  • References: As a general rule, it is better to use the references too often than not enough. In my opinion, the references should be completed. Please add more relevant literature.

Round 2

Reviewer 1 Report

The authors have addressed all the reviewer's comments.

Reviewer 2 Report

This paper has been improved. For that reason my recommendation is- accept in present form.